# Tumor Suppressor Protein p53 and Inhibitor of Apoptosis Proteins in Colorectal Cancer—A Promising Signaling Network for Therapeutic Interventions

**DOI:** 10.3390/cancers13040624

**Published:** 2021-02-04

**Authors:** Ömer Güllülü, Stephanie Hehlgans, Claus Rödel, Emmanouil Fokas, Franz Rödel

**Affiliations:** 1Department of Radiotherapy and Oncology, University Hospital Frankfurt, Theodor-Stern-Kai 7, 60590 Frankfurt am Main, Germany; oemer.guelluelue@kgu.de (Ö.G.); stephanie.hehlgans@kgu.de (S.H.); claus.roedel@kgu.de (C.R.); emmanouil.fokas@kgu.de (E.F.); 2Frankfurt Cancer Institute (FCI), Goethe University Frankfurt, Theodor-Stern-Kai 7, 60590 Frankfurt am Main, Germany; 3German Cancer Research Center (DKFZ), Im Neuenheimer Feld 280, 69120 Heidelberg, Germany; 4German Cancer Consortium (DKTK) partner site, Frankfurt, 60590 Frankfurt am Main, Germany

**Keywords:** cIAP1/2, colorectal carcinoma, inhibitor of apoptosis protein family, Survivin, TP53, XIAP, BRUCE, LIVIN

## Abstract

**Simple Summary:**

Tumor suppressor 53 (p53) is a multifunctional protein that regulates cell cycle, DNA repair, apoptosis and metabolic pathways. In colorectal cancer (CRC), mutations of the gene occur in 60% of patients and are associated with a more aggressive tumor phenotype and resistance to anti-cancer therapy. In addition, inhibitor of apoptosis (IAP) proteins are distinguished biomarkers overexpressed in CRC that impact on a diverse set of signaling pathways associated with the regulation of apoptosis/autophagy, cell migration, cell cycle and DNA damage response. As these mechanisms are further firmly controlled by p53, a transcriptional and post-translational regulation of IAPs by p53 is expected to occur in cancer cells. Here, we aim to review the molecular regulatory mechanisms between IAPs and p53 and discuss the therapeutic potential of targeting their interrelationship by multimodal treatment options.

**Abstract:**

Despite recent advances in the treatment of colorectal cancer (CRC), patient’s individual response and clinical follow-up vary considerably with tumor intrinsic factors to contribute to an enhanced malignancy and therapy resistance. Among these markers, upregulation of members of the inhibitor of apoptosis protein (IAP) family effects on tumorigenesis and radiation- and chemo-resistance by multiple pathways, covering a hampered induction of apoptosis/autophagy, regulation of cell cycle progression and DNA damage response. These mechanisms are tightly controlled by the tumor suppressor p53 and thus transcriptional and post-translational regulation of IAPs by p53 is expected to occur in malignant cells. By this, cellular IAP1/2, X-linked IAP, Survivin, BRUCE and LIVIN expression/activity, as well as their intracellular localization is controlled by p53 in a direct or indirect manner via modulating a multitude of mechanisms. These cover, among others, transcriptional repression and the signal transducer and activator of transcription (STAT)3 pathway. In addition, p53 mutations contribute to deregulated IAP expression and resistance to therapy. This review aims at highlighting the mechanistic and clinical importance of IAP regulation by p53 in CRC and describing potential therapeutic strategies based on this interrelationship.

## 1. Introduction

Colorectal cancer (CRC) accounts for around 10% (more than 1.2 million cases) of annually diagnosed malignancies in the world. It is the fourth most mortal cancer with about 900,000 deaths per year and the incidence is predicted to increase approximately to 2.5 million new cases by 2035 [1]. CRC development is characterized by a multistep process involving a series of histological and morphological changes triggered by a sequential accumulation of specific genomic alterations [2]. Adenomatous polyposis coli (APC) gene mutations occurring in normal colon epithelial cells are among the early incidents of a complex tumorigenesis, resulting in abnormally growing benign precancerous polyps (adenomas and sessile serrated polyps) that, over time acquire the ability to invade the bowel wall and trigger low-grade dysplasia. Successively, promoted by Kirsten rat sarcoma virus (KRAS) oncogene activation and serine/threonine-protein kinase B-Raf (BRAF) mutations along with chromosomal (microsatellite) instabilities, high-grade dysplasia will develop that accelerate transformation to malignant progression and invasive carcinoma by further accumulating p53 mutations. Finally, these local malignancies may acquire the potential to metastasize to local lymph nodes and distant organs [1,3]. TP53 gene mutation frequency is 60% in colorectal cancers with the vast majority of mutations located in the DNA-binding domain of the protein. About 60% of TP53 mutations result in an abrogated function of one allele (loss of heterozygosity); however, this can have a dominant negative effect (DNE) and repress wild-type (wt)-p53 functions. By contrast, gain of function (GOF) mutations may induce tumor initiation and progression as well as cancer stemness, invasion, migration and therapy resistance [4,5,6]. 

In dependence of tumor stage, location and lymph node status, CRC is treated by surgery, neoadjuvant (before surgery) or adjuvant chemotherapy (after surgery) with or without concurrent irradiation. By this, treatment of rectal adenocarcinoma represents a particularly good example for a successful implementation of multimodal concepts in cancer management. The establishment of neoadjuvant therapy based on chemoradiation (CRT) prior to surgical resection was a turning point in the treatment of this entity resulting in substantially reduced local recurrence rates and improved survival by the inclusion of oxaliplatin [7,8]. However, despite identical tumor histology and comparable tumor stages, patient’s response to neoadjuvant CRT ranges from a clinically and pathologically confirmed complete remission in 10–30% of cases to progression under treatment [9]. This variable tumor response further displays a strong prognostic impact and significantly correlates with disease-free (DFS) and overall survival (OS) [10,11], while a comprehensive understanding of the molecular basis that defines the individual therapy response is still at its early stage. Among the molecular tumor determinants associated with carcinogenesis, enhanced proliferation, invasion, migration and resistance to anticancer treatment, members of the inhibitor of apoptosis (IAP) family proteins, most pronounced cellular IAP (cIAP1, cIAP2), X chromosome linked IAP (XIAP) and Survivin, have gained increasing interest [12]. In this review, we aim to illustrate a mechanistic interrelationship between IAPs and p53 that may pave the way to develop new combinational therapies to overcome mutant p53 and IAPs based therapy resistance in CRC.

## 2. Biology and Functions of p53, a Brief Introduction

The p53 protein and poly(ADP-ribose) polymerases (PARPs) are considered to be the “guardians of the genome” due to their role in conserving genetic stability by preventing mutations and mediating tumor suppression via a tightly regulated network in response to stress signals, which results in either cell death or survival [4]. PARP-1, by a direct poly(ADP-ribosyl)ation of the p53 protein results in a nuclear accumulation and transcriptional activation of p21 [13] or is required for ATM-mediated p53 activation and gene expression [14]. In addition, epigenetic repression (p53) and activation (PARP-1) of DNA (cytosine-5)-methyltransferase 1 (DNMT1) activity, a key enzyme implicated in the silencing of DNA repair genes, may represent an indirect interrelationship between both proteins [15].

The major structural part of the p53 protein covers a central DNA-binding domain (DBD), which is connected to the tetramerization domain by a linker region. The regulatory domain is located adjacent to the homo-oligomerization (OD) domain at the protein’s carboxy-terminal end. The vast majority of p53 mutations are located in the DNA-binding region [16]. Transcriptional functions of p53 are mediated by binding to variable consensus sequences in responsive elements in the promoter of target genes. Moreover, p53 also regulates genes partially or completely lacking these consensus sequences dependent on their secondary structure [17]. In addition, p53 directly binds and regulates proteins such as ataxia telangiectasia mutated (ATM) kinase and transcription factors such as Y-box-binding protein (YB-1) [18]. This diversity enables multiple regulatory functions in cellular pathways but needs to be tightly controlled. In that context, a negative feedback loop via murine double minute 2 homologue (MDM2) and MDM4 controls p53-mediated transcriptional and post-transcriptional activity and keeps p53 at low levels under physiological conditions [19,20]. Following DNA damage, p53 is activated by post-translational modifications, e.g., phosphorylation by phosphatidylinositol 3-kinase-related kinase (PIKK)-family members ATM, ataxia telangiectasia and Rad3-related (ATR) and DNA-dependent protein kinase catalytic subunit (DNA-PKcs) or indirect phosphorylation by ATM/ATR/DNA-PKcs substrates checkpoint kinases 1 (CHK1) and CHK2 [19,21]. These modifications result in p53 stabilization, activation and nuclear translocation, followed by p53-mediated transcription of a plethora of target genes, involved in cell cycle regulation, DNA damage repair and apoptosis [22]. 

## 3. Structure and Function of the Inhibitor of Apoptosis Protein Family (IAP) 

The IAP family was first described in 1993 as a class of baculoviral proteins characterized by a functional baculovirus IAP repeat (BIR) domain [23] that prevented apoptosis of insect cells during viral infection [24]. Since their discovery, BIR containing (BIRC) proteins were reported in yeast, insects and mammalians. The human IAP family currently covers eight members, including neuronal apoptosis inhibitory protein (NAIP/BIRC1), cellular IAP1 (cIAP1/BIRC2), cellular IAP2 (cIAP2/BIRC3), X-chromosome-linked IAP (XIAP/BIRC4), Survivin (BIRC5), BIR repeat-containing ubiquitin-conjugating enzyme (BRUCE/Apollon/BIRC6), LIVIN (BIRC7) and human IAP-like 2 (hILP2/BIRC8) [25,26]. The family members differ substantially in protein size and functional domains but share at least one of the family-defining BIR domain facilitating protein-protein interactions with other factors. Additional functional domains include a centrally located ubiquitin associated (UBA) domain present in cIAP1 cIAP2, XIAP and hILP2 to allow these proteins to bind to poly-ubiquitin chains, or a carboxy-terminal localized really interesting new gene (RING) domain conferring ubiquitin ligase activity and mediating signal transduction, protein-protein interactions, and transcription [25]. Further, a caspase activation and recruitment domain (CARD), unique to cIAP1/2, helps to control their ubiquitin ligase activity and stability. In addition, NAIP includes a NAIP-C2TA-HETE-TEP1 nucleotide-binding and oligomerization domain (NACHT) which functions in apoptosis inhibition and major histocompatibility complex (MHC) class II transcriptional activation and a leucine-rich repeat (LRR) domain with a role in signaling pathways of innate immunity and host-pathogen recognition [27]. Finally, Survivin carries an amphipathic α-helical coiled-coil domain at the C-terminus, common in microtubule-associated proteins [28]. 

Although IAPs are primarily considered as sole inhibitors of apoptosis, growing evidence evolves regarding their vital impact as transduction intermediates in a diverse set of signaling pathways associated with the regulation of cell migration, cell cycle and DNA damage response. Some of these functions will briefly be described below and are summarized in Figure 1.

### 3.1. cIAP1 and cIAP2

Primarily, cIAP1 was discovered by its involvement in inflammation/apoptosis signaling interacting with tumor necrosis factor receptor-2 associated factors (TNFR2-TRAFs) via its N-terminal BIR domain [29]. The major biological activities of cIAP1 cover a positive regulation of the canonical transcription factor nuclear factor kappaB (NF-κB) activation pathways. By this, cIAP1 complexes with TRAF2, Src homology 2 domain-containing protein tyrosine phosphatase 1 (SHP1), Src and myeloid differentiation primary response 88 (MyD88) to promote canonical activation of NF-κB [30]. Briefly, in the TNFR1 complex, cIAP1/2 serve as ubiquitin ligases for receptor-interacting serine/threonine-protein kinase 1 (RIPK1), which is needed for TNF-α mediated NF-κB and mitogen-activated protein kinase (MAPK) signaling, gene expression, differentiation, mitosis and inhibition of both, caspase-dependent and -independent cell death. In addition, cIAP1/2 limit the non-canonical NF-κB activation pathway. Here, cIAP1/2 act as ubiquitin ligases that target NF-κB-inducing kinase (NIK) for degradation. Further, upon viral infection, cIAP1/2 ubiquitinate TRAF3/6 which is an essential factor for NF-κB deregulation, while attenuation of cIAP1/2 impedes an antiviral response via inhibition of the virus-triggered activation of NF-κB, interferon regulatory factor 3 (IRF3) and interferon-beta (IFN-β) induction [31]. 

The expression level of cIAP1 is regulated by a variety of transcriptional and post-translational mechanisms including microRNAs (miRNAs) and proteasomal degradation. For instance, microRNA-29c (miR-29c) binds to the 3′-UTR of cIAP1 effectively downregulating its mRNA and protein levels [32]. Further, downregulation of ubiquitin thioesterase OTU domain ubiquitin aldehyde binding 1 (OTUB1) enhances the degradation of cIAP1 and inhibits the TNF-related weak inducer of apoptosis (TWEAK)-induced MAPK and NF-κB pathway [33]. Upon genotoxic stress, a bilateral cell death regulation involving the ripoptosome (RIP1/3-FADD-Caspase-8-c-FLIP) complex is reported: cIAP1 and cIAP2 directly ubiquitinate RIP1 which associates with the pro-survival transforming growth factor beta-activated kinase 1 (TAK1) and triggers proteolytic degradation of the ripoptosome complex while attenuation of IAPs allows the proper formation of the complex [34]. 

Beside its function in concert with cIAP1, cIAP2 is involved in response to metal stress, DNA repair and together with XIAP and Survivin in exosomal secretion [35,36,37]. The latter may not only serve as warning signals, but may also play a role in providing protection to the cancer cells against potential dangers in the tumor microenvironment [38]. 

As a response to histone deacetylase (HDAC) inhibitor Panobinostat treatment, cIAP2 exhibits the highest upregulation in line with a decreased level of DNA double-strand break (DSB) repair protein meiotic recombination 11 homolog (MRE11). Moreover, cIAP2 directly interacts with MRE11, promotes its ubiquitination and directs it to degradation that in turn delays DNA DSB repair resulting in increased radiation sensitivity [35]. 

### 3.2. XIAP

Human XIAP was initially discovered as an IAP-like apoptosis inhibitor protein by its homology to baculovirus IAP genes [39]. XIAP is an archetypical IAP protein that, in contrast to other family members, inhibits the active catalytic sites of caspases-3 and caspases-7 in a direct manner and interferes with the dimerization and activation of caspase-9. This prevents their downstream effector functions, including the release of mitochondrial IAP antagonists such as second mitochondrial activator of caspases (SMAC/Diablo) and the serine peptidase HtrA2/Omi, that bind XIAP’s BIR domains, releasing active caspases into the cytosol [40]. In addition, recent studies presented XIAP as a multifunctional protein involved in cellular and metabolic regulatory circuits such as invasion, migration, necroptosis, oxidative stress, inflammasome formation and autophagy [41,42,43,44,45,46,47,48,49,50]. XIAP´s BIR domain 1 mediates activation of stress-responsive signaling pathways, such as Jun Kinase (JNK) [51] and MAPK phosphorylation cascade that in turn activate NF-κB. XIAP also activates NF-κB by promoting the translocation of the p65 subunit to the nucleus and by degradation of the NF-κB inhibitor IκB [52]. XIAP empowers interleukin-17 mediated NF-κB activation and caspase-3 inhibition that drives colon tumor formation [44]. In line with Survivin, XIAP represents a radiation resistance factor and attenuation of the protein triggers radiation response in CRC cancer cell lines [53,54].

### 3.3. Survivin

Survivin is among the most studied members of the IAP family. The protein was discovered in the late nineties as its smallest member involved in fetal development and cancer progression. Survivin is downregulated in most terminally differentiated cells and re-expressed in the majority of solid and liquid human tumors investigated [55,56]. Survivin is a prime example of a multifunctional protein involved in a variety of regulatory circuits in tumor cells [37,57]. By this, a present conception is that most IAPs, except for XIAP, block apoptosis by mechanisms other than direct caspase inhibition [58], but via cooperative interactions with other partners. Thus, an association of Survivin with hepatitis B X-interacting protein (HBXIP) and/or XIAP inhibits caspases, while binding to SMAC/Diablo counteracts this activity. Moreover, Survivin is expressed in a cell cycle regulated manner participating in cell division as an interactor of chromosomal passenger complex (CPC) proteins INCENP, Borealin and Aurora-B [59,60]. In malignant cells, however, Survivin is regulated independently of mitosis by a variety of oncogenic pathways. Further, Survivin is a predominantly nucleocytoplasmic protein; however, shuttling to or from other compartments like the nucleus is mediated by Exportin-1, irradiation and post-translational regulations such as homodimerization and acetylation of residue K129 [61,62]. Survivin is subjected to multiple post-translational modifications that mostly are decision-makers on its functions and fate of the host. For instance, phosphorylation of residue T34 by p34(cdc2)-cyclin B1 facilitates proper Survivin-caspase-9 interaction that results in inhibition of apoptosis [63]. In addition, Survivin is a radiation-inducible factor mediating the cellular radiation response in a multitude of tumors including colorectal cancer [64,65,66]. By this, Survivin accumulates in the nucleus and interacts with a prime non-homologous end joining repair factor DNA-dependent protein kinase (DNA-PKcs) [67,68]. Survivin forms a heterotetramer complex with DNA-PKcs that results in a conformational change on the DNA-PKcs phosphoinositide 3-kinase domain with enhanced enzymatic activity and detection of differentially abundant phosphopeptides and proteins implicated in the DNA damage response [69]. 

### 3.4. BRUCE/Apollon

With a molecular weight of 528 kDa, BRUCE/Apollon is a huge E3 ubiquitin transferase whose mutation causes embryonic lethality [70,71]. In concert with other IAPs, one of the main functions of BRUCE is inhibition of apoptosis [70]. BRUCE binds and ubiquitinates SMAC/Diablo, caspase-9 and mitochondrial serine peptidase HtrA2/Omi to prevent apoptosis by facilitating their proteasomal degradation [72,73]. Early in mitosis, BRUCE binds the anaphase-promoting complex/cyclosome (APC/C) and enables the degradation of Cyclin A by ubiquitination independent of cyclin-dependent kinases (CDKs) [74]. However, in the late phase of cytokinesis, BRUCE is an essential component of the midbody ring and the tubular recycling system under the regulation of mitotic kinesin-like protein-1 to regulate cytokinetic abscission [75]. Upon DNA damage, BRUCE acts as a scaffold to form a complex with ubiquitin-specific peptidase 8 (USP8) and breast cancer susceptibility gene C terminus-repeat inhibitor of human telomerase repeat transcriptase expression-1 (BRIT1) at the DSBs, which is vital for the formation of BRIT1 DNA damage foci [76]. BRUCE further regulates ATR-directed signaling pathways in DNA replication stress via interaction with pre-mRNA-processing factor 19, while depletion of BRUCE causes a stalled DNA replication, prevents the activation of ATR and inhibits the phosphorylation of CHK1 and replication protein A [77].

### 3.5. LIVIN

LIVIN (37/39 kDa) plays a key role in a multitude of cellular mechanisms and stress responses including radiation response, invasion, hypoxia-resistance and autophagy [78,79,80,81,82]. LIVIN inhibits apoptosis by binding both, caspase-3/7 as well as TNFα-induced DEVD-like caspase. In addition, LIVIN indirectly inhibits caspase-9 via apoptotic protease activating factor-1 (Apaf-1) [83]. However, reports on a bidirectional regulation between caspases and LIVIN revealed that the truncation of LIVIN (28/30 kDa) by caspase-3/7 transforms it to a pro-apoptotic protein [84,85]. Chemosensitivity studies in colon cancer cells further revealed LIVIN as a drug resistance gene against etoposide (VP-16) and 5-fluorouracil (5-FU) [86], while attenuation of the protein significantly decreases the size of colon cancer xenograft tumors [86,87]. Upon irradiation, LIVIN overexpression is associated with cellular radioresistance, whereas attenuation of LIVIN decreases radiation-induced cell invasion ability and enhances radiation response [78,79]. 

### 3.6. NAIP and hILP-2

NAIP was first discovered as spinal muscular atrophy (SMA) related gene whose deletion or mutation was restricted to SMA patients [88]. However, comprehensive studies over the last years indicate the relevance of NAIP in a variety of different molecular mechanisms and diseases such as cytokinesis, inflammasome formation, amyloid-β toxicity, amyotrophic lateral sclerosis (ALS) and Parkinson’s disease [89,90,91,92,93]. The least-studied member of IAP family, hILP-2 is discovered as a protein owing a high sequence homology to XIAP but no inhibitory effect on TNF-mediated apoptosis. Nevertheless, it can inhibit Bcl-2-associated X protein (Bax) or caspase-9 and Apaf-1 triggered apoptosis. Moreover, hILP-2 directly interacts with cleaved caspase-9 [94] and attenuation of hILP-2 triggers apoptosis and inhibits migration [95].

## 4. Prognostic Relevance of IAP Expression in CRC

By using a tissue microarray technology, immunoexpression of cIAP1, cIAP2, XIAP, Survivin and their antagonist SMAC/Diablo were evaluated in the treatment of naive colorectal carcinoma and expression levels were correlated with the levels of apoptosis, cellular proliferation and patient’s prognosis [96]. Particularly, a low cIAP1 immunodetection in malignant tissue was correlated with a significantly lowered patient survival. According to a report by Krajewska et al., an elevated expression of cIAP2 in stage II CRC cases significantly correlated with an impaired overall survival (OS) in both uni- and multivariate analyses [97] and is considered to be a predictive marker for the sensitivity to the chemotherapeutic drug 5-fluorouracil [98]. Furthermore, other studies indicate higher levels of XIAP expression to correlate with venous invasion, Duke’s staging, tumor differentiation and multivariate analysis further proving XIAP to be an independent prognostic factor for an impaired DFS and OS [97,99]. 

In addition, Survivin has consistently been demonstrated to be overexpressed in solid human tumors and significantly correlates with tumor onset, more aggressive and advanced pathologic features, metastasis and worse prognosis as well as impaired patient’s survival [25,100,101]. In line with that, Survivin is involved in early development of colorectal cancer [102]. In detail, transcription factors of the T-cell factor (TCF)/beta-catenin family-mediated increased Survivin expression imposes a stem cell-like phenotype in colonic crypt epithelial cells coupling enhanced cell proliferation with resistance to apoptosis and the molecular pathogenesis of colorectal cancer. In addition, activated signal transducer and activator of transcription 3 (STAT3) participates in early stage of colon cancer progression by upregulation of the stem cell marker CD133 that in turn induces Survivin expression [103]. Moreover, Survivin expression increases during the adenoma-carcinoma sequence and is maintained throughout the progression of disease [104]. Survivin overexpression is significantly associated with primary tumor sites, lymph node metastasis and advanced III/IV stages and is an independent prognostic factor for both DFS and OS in multivariate analysis [105]. Notably, a meta-analysis on a total of 1784 patients from 14 studies confirmed Survivin overexpression in patients with CRC to be significantly associated with poor DFS and OS [106]. Concerning a predictive relevance, a failure of downregulation of Survivin from pretreatment biopsies to corresponding posttreatment resection specimens after neoadjuvant radiochemotherapy in rectal cancer was associated with development of distant metastases and correlated significantly with DFS and cancer-specific survival [107,108]. These findings may be explained on a molecular level as Survivin orchestrates NF-κB dependent expression of fibronectin, integrin signaling, activation of focal adhesion kinase (FAK) and Src, and upregulation of v-akt murine thymoma viral oncogene homolog (AKT) pathway to mediate tumor cell migration and metastatic dissemination [109]. Finally, a high cytokeratin-20 and Survivin expression in blood circulating tumor cells predict inferior OS in metastatic CRC patients receiving various chemotherapy regimens [110].

As reported before for XIAP and Survivin, compared to normal mucosa and non-metastatic lymph node tissues, LIVIN expression is also significantly overexpressed in CRC and correlates with poor patient survival [111], tumor stage, lymphovascular invasion and lymph node metastasis [112] that may originate from triggering NF-κB activation and its downstream targets fibronectin and chemokine (C-X-C motif) receptor 4 (CXCR4) as reported for prostate cancer invasion [80]. Finally, BRUCE overexpression is a prognostic marker for colorectal cancer, which correlated with tumor size and invasion depth and was significantly associated with worse OS and shorter DFS in a cohort of 126 patients [113,114]. Concordantly, a large-scale proteomics study of colon spheres enriched with colon cancer stem cells identifies BRUCE as a highly upregulated therapeutic target [115].

In summary, due to their prognostic and predictive relevance along with a prominent role at disparate cellular networks on tumor cell apoptosis, invasion and metastases, IAPs are considered to represent valuable oncotherapeutic targets with the inhibitory approaches (e.g., antisense oligonucleotides, small molecules and immunotargeting) about to enter clinical evaluation [116]. 

## 5. Molecular Regulation of IAPs by p53 

The first report on a putative functional interrelationship between IAPs and p53 arises from an inverse immunohistochemical correlation signature of Survivin and p53 expression in gastric carcinoma [117]. An inverse correlation was subsequently confirmed *in vivo* and *in vitro* in additional tumor entities including breast, ovarian and lung carcinoma cell lines [118,119,120]. More recent findings further strengthen the clinical relevance of an IAP-p53 interrelationship. For instance, a clinical study assessing the gene expression levels in tumor biopsies of colon cancer patients revealed a significant correlation between the gene expression levels of LIVIN and p53. The correlation covers the upregulation of LIVIN and downregulation of p53 which is highly associated with aggressive tumor growth and metastatic spread [121]. P53’s main physiological function is to regulate the genes that control apoptosis [19]. Functionally, Survivin is an inhibitor of apoptosis protein, thus the repression of Survivin by p53 constitutes a mechanism that enables tumor cells to execute apoptosis upon induction by apoptotic stimuli. Indeed, Mirza et al. were the first to report a direct link between Survivin and wt-p53 that contributes to cancer progression [119]. On a functional level, transcriptional repression of Survivin expression is mediated by wt-p53 binding to the promoter region, while transcription factor E2F binds to a comparable promoter binding region and transactivates Survivin expression in the absence of p53 [120,122]. The mechanisms of the transcriptional repression are not fully elucidated to date and may further include DNA methylation and modification of chromatin structure accessibility within the Survivin promoter region [119]. Accordingly, the recruitment of histone deacetylase (HDAC) by p53 to the Survivin promoter is involved in the inhibition of gene transcription [120]. In concordance with the previous finding, inhibition of HDAC2 by siRNA or treatment with deacetylase inhibitor suberoylanilide hydroxamic acid (SAHA) triggers the proteasomal degradation of MDM2 that upregulates p53 and results in a suppression of Survivin [123]. Further, the Survivin promoter contains a canonical CpG island and is hypermethylated in malignant cells that prevents p53 binding and results in a high Survivin expression, while Decitabine-induced DNA demethylation promotes a p53-dependent downregulation of Survivin [124]. Moreover, Zhu et al. identified a regulatory pathway for the expression of Survivin under the control of Kruppel-like factor 5 (KLF5) and p53. KLF5 directly binds to multiple GT-boxes in the core Survivin promoter to strongly induce its transcriptional expression; likewise, KLF5 binds to p53 that abrogates the repression of Survivin [125]. Other investigations, however, did not confirm that p53 could physically interact with the Survivin promoter and indicated an indirect interrelationship. For instance, adenovirus E1B-55K protein is involved in indirect p53-mediated repression of Survivin by interfering with the Sin3 core repressor complex [119,126]. More recent reports support indirect repression mechanisms, including p53-dependent upregulation of miRNAs in CRC cells [127,128]. P53 interacts with Drosha miRNA processing complex and DEAD-box RNA helicase p68 (DDX5) and modulates miRNA biogenesis. In response to DNA damage, p53 regulates the post-transcriptional maturation of several miRNAs including miR-15a and miR-16. According to an *in vitro* study, overexpression of miR-16 significantly enhances apoptosis in HCT116 cells [127]. Notably, high expression of both miR-15a and miR-16 correlated with better DFS and OS in colorectal cancer patients [129,130]. Furthermore, DNA damaging agent Bleomycin induces p53 expression and induction of miR-15a which in turn targets NAIP and decreases its mRNA and protein expression levels [131]. Upon 5-FU treatment of colorectal cancer cells, miR-96 gets upregulated which triggers the downregulation of XIAP and p53 stability regulator ubiquitin conjugating enzyme E2N (UBE2N) resulting in the stimulation of apoptosis [132]. Finally, miR-192-5p and miR-215 constitute p53-responsive miRNAs. Upon knockdown of p53, expression of both miRNAs was abrogated in non-small cell lung cancer and target analysis revealed XIAP as a transcriptional target [133].

P53 is an upstream transcriptional regulator of cyclin-dependent kinase inhibitor p21 (WAF1/CIP1). Survivin acts as a transcriptional repressor/cofactor of p21 mRNA and protein expression via directly interacting with p53 on the p53-binding sites of the p21 promotor, while C84A mutation or knockdown of Survivin reversed this regulatory pathway [134]. Moreover, Survivin ablation has a feedback regulatory function on p53 which stimulates its transcriptional activation and enables the expression of p21 [135]. Glycogen synthase kinase-3beta (GSK-3β) induces nuclear accumulation of Survivin [136], while a GSK-3β dominant-negative mutant or siRNA mediated silencing induces stability and activation of p53 that triggers p21 expression and results in the repression of Survivin [137]. On a molecular level, p53-dependent gene repression by p21 is mediated by a promoter binding of transcription factor Myb-related protein B (B-MYB) mediated switch to binding transcription factor E2F4 and p130 to the dimerization partner, RB-like, E2F and multi-vulval class B (DREAM) complex onto the Survivin promoter [138,139]

Stimulation of IAPs-p53 signaling pathways is mediated by a wide range of external/internal sources, including elevated temperature, metal stress, viral infection and DNA damaging stress such as treatment with chemotherapeutic drugs and ionizing irradiation [140,141,142,143,144,145,146,147]. In Etoposide-dependent induction of apoptosis, p53 induces the expression of HtrA2/Omi which cleaves and inactivates cIAP1. The cleavage process of cIAP1 is caspase-independent while the serine protease inhibitor 4-(2-Aminoethyl)benzolsulfonylfluoride (AEBSF) inhibits this apoptosis induction [140]. 

The cellular stress response is highly related with and coordinated by autophagy which is a mechanism of cellular homeostasis that regulates the starvation response and degradation of damaged molecules and organelles via autophagosome formation. However, autophagy dysfunction causes tumorigenesis and promotes metastasis. Essential autophagy genes such as microtubule-associated protein 1A/1B-light chain 3, Beclin-1, and autophagy protein-5 are considered as markers for staging and survival prognosis in colorectal cancers [148]. P53 is a well-recognized negative regulator of autophagy [149]. In HCT116 colorectal cancer cells, XIAP knockdown/knockout markedly promotes MDM2 levels that induces the degradation of p53 while phospho-S87 XIAP interacts and facilities the degradation of MDM2 that maintain high levels of cytosolic p53. In concordance, S87A mutation of XIAP hampers its anti-autophagy activity [49]. Nevertheless, in stressed conditions, inhibition of the PI3K/Akt pathway results in a dephosphorylation of S87 of XIAP and hampers the XIAP-mediated MDM2 degradation which results in a predominant nuclear p53 accumulation and proper facilitation of autophagy [49]. Notably, the severity of cellular stress further initiates opposite IAP-p53 regulatory mechanisms. Arsenite treatment with low concentrations of the drug upregulates a cascade including activation of extracellular signal-regulated kinases (ERKs) which in turn trigger NF-κB activation. Upon NF-κB activation mitochondrial 70 kDa heat shock protein family A (Hsp70) member 9 (HSPA9) gets upregulated and hereby downregulates p53 that triggers the upregulation of Survivin and induction of cell proliferation. By contrast, upon treatment with high concentrations of arsenite, c-Jun N-terminal kinases (JNKs) get activated which inhibits the degradation of p53 by MDM2. Upon p53 stabilization, Survivin is downregulated, resulting in the execution of apoptosis [147]. For a further bidirectional regulation between IAPs and p53, E2/E3 ubiquitin ligase BRUCE has been identified as an upstream regulator of p53. Mechanistic studies revealed that BRUCE directly binds to p53 and facilitates its proteosomal degradation [114]. Antisense attenuation or C-terminal deletion (∆UBC domain) of BRUCE stabilizes the p53 protein and directs it to the nucleus that results in a transcriptional upregulation of pro-apoptotic genes BAX, Bcl2-antagonist/killer 1 (BAK) and p53-inducible protein with a death domain (PIDD) and activation of caspase-3 that ends up in a G1/S arrest and stimulation of mitochondrial apoptosis [114,150,151]. In addition, Zinc oxide nanoparticle and copper complex-induced stress triggers stability and activation of p53 via phosphorylation on residues S15, S46 and S392 which decreases the expression of Survivin, XIAP, cIAP1 and LIVIN [145,152]. Furthermore, heat stress at 42 °C directs p53 predominantly to the nucleus and activates the protein via S15 phosphorylation. Activation of p53 induces the upregulation of pro-apoptotic genes such as BAX, p53 upregulated modulator of apoptosis (PUMA) and p21 as well as downregulation of XIAP that promotes mitochondrial SMAC release and induction of caspase-3 expression [153]. 

Topoisomerase I (Topo I) is an essential enzyme involved in DNA replication and transcription which relaxes supercoiled DNA by nicking one strand of double-stranded (ds)-DNA. DNA nicking generates reversible cleavage complexes, however, they may cause irreversible DNA lesions including mismatches and breaks that trigger DNA damage response mechanisms [154]. DNA damage caused by the combined treatment of proteasome inhibitor PS-341 and Topo I inhibitor SN-38 stabilizes p53 and downregulates Survivin, triggering p53-dependent apoptosis [155]. Likewise, the combination of Topo I inhibitor Irinotecan and Survivin dimerization inhibitor LLP3 results in an upregulation of X-linked inhibitor of apoptosis factor 1 (XAF-1) and downregulation of Survivin in p53-mutated CRC cell lines [156]. In line with that, a recent study exploring mechanisms to overcome drug resistance by applying a combination of Irinotecan and IAP small molecule antagonist BV6 revealed a chemosensitization by cIAP1/2 degradation and caspase-8/9 activation in mismatch repair (MMR)-proficient but not in MMR-deficient p53 mutated colorectal cancer cells and organoids [157].

Retinoblastoma protein (RB) is reported to downregulate the expression of cIAP1 and cIAP2 in p53-deficient conditions while attenuation of p53 and RB initiates transcription factor E2F mediated transactivation of cIAP1 and cIAP2 genes [158]. On the contrary, cIAP2 knockdown triggers an alternative NF-κB pathway (Inhibitor of nuclear factor kappa-B kinase subunit alpha (IKKα)-mediated) which activates MDM2 via SUMOylation and S166 phosphorylation that results in MDM2-dependent degradation of p53. IKKα forms a complex with the SUMO-E3 ligase protein inhibitor of activated STAT 1 (PIAS1) that holds the ligase in an inactive state. Upon cIAP2 knockdown, activated IKKα phosphorylates S90 residue of PIAS1 and disrupts this complex. Released PIAS1 then SUMOylates MDM2 [159].

STAT3 is a multifunctional transcription factor with an essential role in colon cancer progression and inflammation. By this, activated STAT3 may participate in tumor progression through increasing CD133/Survivin expression in early stage of colon cancer development [160]. Upon irradiation, increased expression of cyclooxygenase-2 (COX2) induces prostaglandin receptor E2 (PGE2) expression that induces STAT3 pathway and results in an upregulation of Survivin [161,162]. Attenuation of the STAT3 pathway in HCT116 and SW480 colorectal cancer cells, by contrast, downregulates Survivin and upregulates p53 and caspase-3 [160]. Further, ionizing radiation has an acute proteome acetylation effect [163], which triggers the CREB-binding protein (CBP)-dependent acetylation of Survivin on lysine 129 and directs its nuclear localization [61]. Notably, acetylated nuclear Survivin directly binds to the N-terminal transcriptional activation domain of STAT3 and represses transactivation of target gene promoters [61]. IAP-p53 interacting pathways are schematically summarized in Figure 2.

## 6. Clinical Treatment Potential by IAPs and p53

As reviewed above, interfering with the multifaceted interrelationship between IAPs and p53 holds a promise to overcome treatment resistance and sensitize cancer cells with particularly p53 mutant phenotypes to apoptosis. Indeed, there are some recent preclinical and clinical approaches to target IAPSs alone or in combination with p53. Strategies to target IAPs comprise an impressive spectrum of antagonists covering SMAC mimetics, [164,165,166,167,168], RNA interference (small interfering RNA, siRNA) [112,169,170,171,172], small molecule transcriptional inhibitors (YM155) [173], Survivin-T43A mutant RNA therapy [174], peptides targeting Survivin-XIAP complexes (Sur-X) [175] and Survivin dimerization modulators (LLP3) [156]. As depicted in more detail in Table 1, the modes of action of these inhibitors include activation of intrinsic and extrinsic (TNF-α, Tumor necrosis factor-related apoptosis-inducing ligand, Trail) apoptosis, suppression of tumor cell migration and invasion, inhibition of proliferation and sensitization to radiation and chemotherapy.

MX69 is a recently established inhibitor of MDM2 protein-XIAP RNA interaction mediating MDM2 degradation and activation of wt-p53 fostering apoptosis *in vitro* and inhibition of cancer cell proliferation in a mouse xenograft model [176]. Another approach combining MDM2 inhibition via Nutlin-3a and XIAP inhibition by small molecule antagonists synergistically resulted in an elevated level of apoptosis compared to single targeting in acute myeloid leukemia OCI-AML3 and MOLM13 cancer cells [177]. Other preclinical approaches are mainly concentrating on targeting the p53/Survivin axis. For that purpose, several drugs including statin Lovastatin, small molecular compound CT-1042 and natural compound Phoyunnanin-E show promising findings particularly on the transcriptional suppression of Survivin, activation of p53 and sensitization of treatment resistant cancer cells to apoptosis [178,179,180] (Table 1).

Clinical phase I/phase II trials currently recruiting, at least in part, patients with CRC include Survivin epitope peptide vaccines (www.clinicaltrials.gov (accessed on 5 January 2021); NCT00108875), combinations of SMAC mimetics and checkpoint (PD-1) inhibitor Pembrolizumab (NCT02587962, NCT03871959) or standard regimes of chemotherapy (NCT01188499). In addition, there are two approaches completed in phase I and II clinical trials via autologous dendritic cells pulsed with human Survivin, telomerase (hTERT) and p53-derived peptides to induce an anti-tumor immune response in patients with HLA-A2 positive metastatic breast cancer and melanoma (NCT00197912 and NCT00978913). Further, the six-month survival was increased when dendritic cell (DC) therapy was combined with a nonsteroidal anti-inflammatory COX-2 inhibitor and immune suppressor Cyclophosphamide [181]. Interestingly, COX-2 inhibition indirectly suppresses the expression of Survivin via modulating the epidermal growth factor receptor (EGFR)/STAT3 pathway [162]. Published data further indicate a clinical improvement and enhanced levels of peptide-specific cytotoxic T lymphocytes following combined HLA-A24-restricted antigenic peptide Survivin-2B80-88 vaccine and incomplete Freund’s adjuvant and interferon (IFN)alpha treatment [182]. Clinical studies targeting IAPs and p53 and therapeutic treatment strategies targeting IAPs are summarized in Table 2 and Figure 3, respectively.

Challenges for interfering with IAPs in therapeutic approaches mainly comprise the stratification of patients according to their responsiveness, requiring the identification of predictive biomarkers. The increase of apoptotic and necroptotic markers, e.g., caspase-3, caspase-7 or TNF, which has been shown to be important for the activity of IAP antagonists, might serve as such markers [183,184]. Further, since inhibition of IAPs in combination with chemotherapeutics, kinase inhibitors, radiotherapy or immunotherapy seems to be far more promising, it will be of importance to carefully explore the mode of action of these multimodal options to optimize the therapeutic window for the safe and efficient application and to prevent adverse side effects, e.g., cytokine release syndrome in patients treated with the SMAC mimetic LCL161 [185].

## 7. Conclusions

In recent years our understanding of the functions of IAPs has expanded beyond their ability to interfere and inhibit caspase activity and cell death, with newly identified properties, especially in malignant cells. The number of their cellular roles continues to expand as studies implicate IAP involvement in a growing number of signaling and regulatory cascades, including DNA damage response, invasion and metastatic processes. Moreover, due to their early involvement in tumorigenesis and prognostic/predictive relevance in a multitude of malignancies, IAPs cover a group of oncotherapeutic molecular targets to overcome treatment resistance and to improve the effectiveness of chemo- and radiation therapy. As depicted in this review, IAPs further experience a huge array of interrelationships with the tumor suppressor p53, a marker with pivotal importance in tumor development, maintenance and therapy response. Although combined modality options are at an early stage of development, a growing knowledge on physiological and pathophysiological interconnections may pave the way to develop innovative concepts based on IAP-binding/regulatory proteins such as SMAC mimetics and activators/inhibitors of wild type and mutated p53, respectively. 

## Figures and Tables

**Figure 1 cancers-13-00624-f001:**
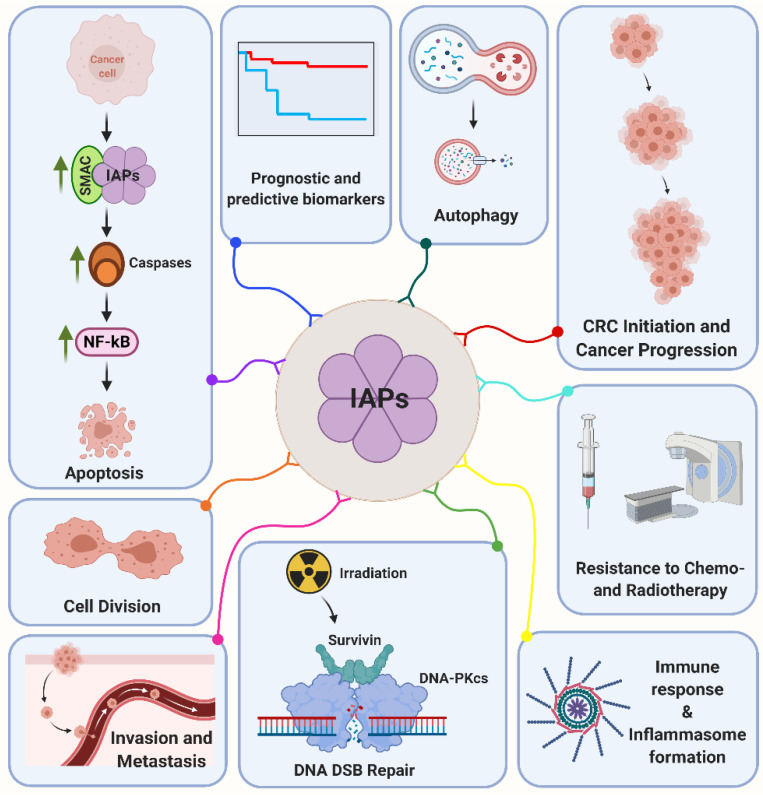
Inhibitor of apoptosis proteins (IAPs) are multifunctional proteins that regulate a variety of key cellular mechanisms such as apoptosis, cell division, invasion and metastasis, autophagy, DNA double-strand break (DSB) repair, cancer progression, immune response and inflammasome formation. Moreover, IAPs are associated with radiation and chemotherapy resistance and are considered to be valuable prognostic and predictive biomarkers in colorectal cancer (CRC). Please see the text for a more detailed discussion. Abbreviations: DNA-PKcs, DNA-dependent protein kinase, catalytic subunit; NF-κB, nuclear factor kappa B; SMAC, second mitochondrial activator of caspases.

**Figure 2 cancers-13-00624-f002:**
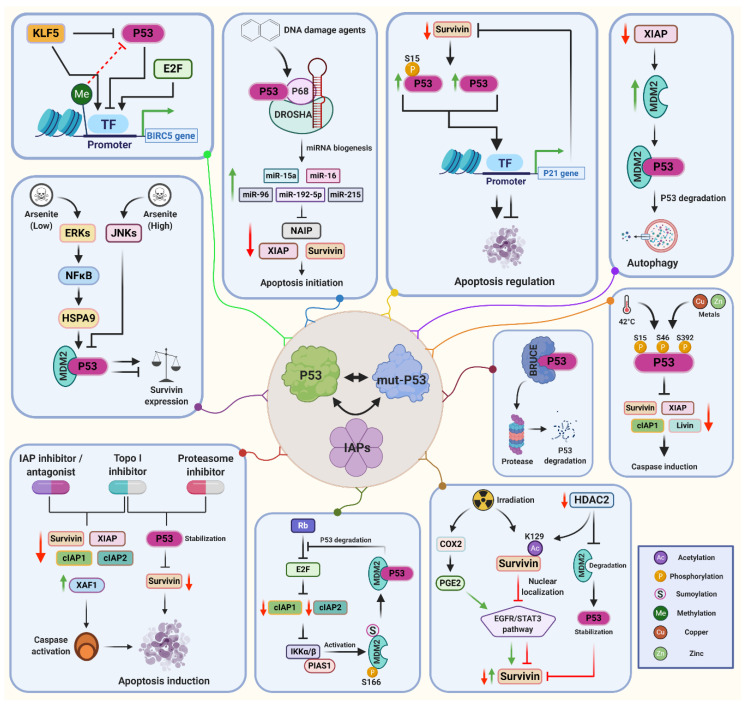
The IAP-p53 axis is an essential regulator of a multitude of cellular pathways. P53 is a key transcriptional suppressor of Survivin expression. Upon varying levels of different stress sources such as DNA damage agents, arsenite, heat and metal stress, activated p53 triggers caspase induction and apoptosis initiation via p21 expression and IAP-targeting of micro(mi)RNA biogenesis. Murine double minute 2 homologue (MDM2) as well as BRUCE function as upstream regulators of p53, whose binding arrests and directs it to targeted degradation. In addition, the IAP-p53 axis is also involved in the mediation of autophagy, epidermal growth factor receptor (EGFR)/ signal transducer and activator of transcription 3 (STAT3) pathway, NF-κB and alternative NF-κB pathways (Inhibitor of nuclear factor kappa-B kinase subunit alpha (IKKα)-mediated) under the regulation of MDM2 and histone deacetylase 2 (HDAC2). Details are given in the text. Abbreviations: BIRC5, BIR containing 5; cIAP1, cellular IAP1; cIAP2, cellular IAP2; Cox-2, cyclooxygenase-2; ERK, extracellular signal-regulated kinase; E2F, transcription factor E2F; HSPa9, heat shock protein family A (Hsp70) member 9; JNK, Jun kinase; KLF5, Kruppel-like factor 5; mut-P53, mutated p53; NAIP, neuronal apoptosis inhibitory protein; PGE2, prostaglandin receptor E2; PIAS1, protein inhibitor of activated STAT 1; Rb, Retinoblastoma protein; TF, transcription factor; Topo I, topoisomerase I; XAF1, X-linked inhibitor of apoptosis factor 1; XIAP, X-linked inhibitor of apoptosis protein.

**Figure 3 cancers-13-00624-f003:**
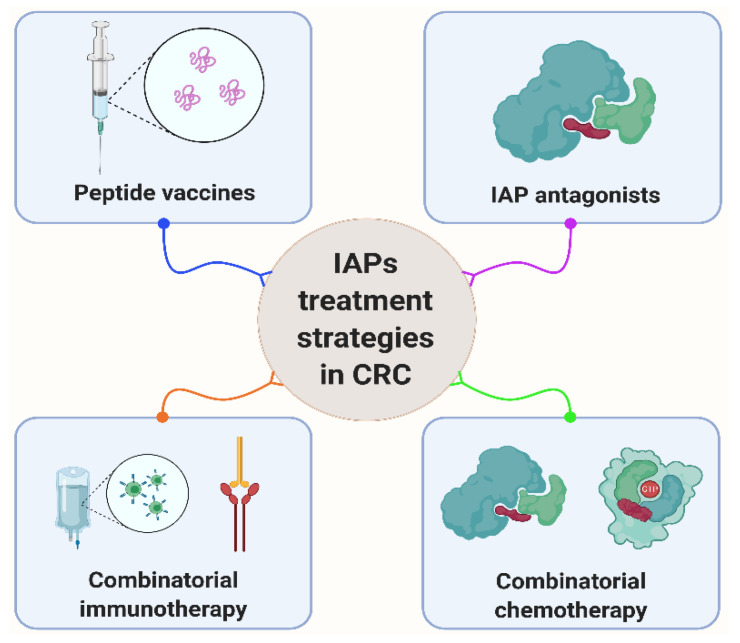
Therapeutic strategies that target IAPs in CRC. Current approaches include peptide vaccines, IAP antagonists and combinatorial treatment strategies further empowering with immuno-/chemotherapeutics such as anti-PD1 antibodies. Further details are given in the text and in Table 2.

**Table 1 cancers-13-00624-t001:** Recent preclinical studies targeting inhibitor of apoptosis proteins (IAPs) in colorectal cancer (CRC).

Target	Inhibitor	Mode of Action	Reference
**Targeting IAPs**			
cIAP1/2 XIAP	SMAC mimetics: BV6, Birinapant AT-406, Tolinapant	BV6 enhances the cellular radiosensitivity and the number of radiation-induced DNA damage foci in CRC cells. Birinapant alone or in combination with AT-406 increases apoptosis upon oxaliplatin/5-FU treatment. Tolinapant facilitates the activation of extrinsic apoptosis pathway by stimulating TNF-α and TRAIL and caspase activation	[164,165,166,167]
cIAP1/2	siRNA	cIAP1/2 silencing chemosensitizes colorectal cancer cells and triggers caspase-8 activation and apoptosis	[169]
XIAP	Sur-X peptide inhibitor, siRNA, BV6, Mithramycin-A	Sur-X peptide inhibitor prevents the Survivin-XIAP interaction and inhibits their antiapoptotic and prometastatic activities; attenuation of XIAP by siRNA, Mithramycin-A or BV6 overcomes TRAIL-dependent apoptosis resistance in CRC cells	[168,172,175]
Survivin	YM155, LLP3 protein inhibitor, Sur-X peptide inhibitor, Survivin-T34A gene therapy, EpCAM-aptamer-guided survivin RNAi	YM155 treatment induces apoptosis by modulating ER-stress mediated apoptosis signaling. LLP3 drives the proteolytic degradation of Survivin and sensitizes CRC cells when combined with irinotecan (Topoisomerase inhibitor). Survivin-T34A mRNA encapsulated with a liposome-protamine lipoplex exhibits superior antitumor effect in a CRC mouse model. The aptamer-guided survivin RNAi enhances chemosensitivity, increases apoptosis, inhibits tumor growth in CRC stem cells and improves survival in xenograft mice	[156,171,173,174,175]
Livin	siRNA	Silencing of Livin enhances chemosensitivity to 5-FU in CRC cells, regulates crosstalk between apoptosis and autophagy and supresses tumor cell migration and invasion	[112,170]
**Targeting IAPs and p53**			
MDM2 XIAP	MX69 (dual MDM2 protein and XIAP RNA inhibitor)	MX69 triggers the downregulation of XIAP and degradation of MDM2 enabling activation of p53, induction of apoptosis and inhibition of cell proliferation *in vivo*	[176]
MDM2 XIAP	Nutlin-3 and XIAP antisense oligo- nucleotides	Nutlin-3 induces MDM2 degradation and activation of p53 that results in apoptosis induction in synergy with XIAP attenuation by antisense oligonucleotides	[177]
Survivin-p53 axis	Lovastatin (statin) CT-1042 (small molecule), Phoyunnanin-E	Lovastatin, CT-1042 and Phoyunnanin-E are involved in suppression of Survivin, activation of p53 and sensitization of cancer cells to apoptosis	[178,179,180]

Abbreviations: MDM2, Murine double minute 2 homologue; RNAi, RNA interference; siRNA, small interfering RNA; Trail, Tumor necrosis factor-related apoptosis-inducing ligand.

**Table 2 cancers-13-00624-t002:** Clinical studies targeting IAPs and p53.

Identifier	Disease	Treatment	Purpose	Outcome Measures
NCT00108875	melanoma, pancreatic and cervical cancer, CRC	Survivin epitope peptide vaccine	to evaluate the safety, immunological response and clinical outcome	PFS, OS, best response, immunological response
NCT02587962	Phase 1: solid tumors Phase 2: ovarian and cervical cancer, CRC	Birinapant (SMAC-mimetic), Pembrolizumab (PD-1 inhibitor)	to evaluate the safety, tolerability, pharmaco-dynamics and efficacy of combined modality treatment	Blood pressure, electro- cardiogram, enzymes, hemoglobin, physical exam, overall response
NCT03871959	non-MSI-high advanced or metastatic pancreatic, colon and rectum cancer	DEBIO1143 (SMAC-mimetic) Pembrolizumab	to determine the MTD, the recomm. dose for a phase 2 trial and to evaluate efficacy	MTD and the dose for phase 2 of Debio1143 when combined with a fixed dose of Pembrolizumab, duration of response, clinical benefit, tumor response efficacy, PFS
NCT01188499	advanced or metastatic tumors including CRC	Birinapant TL32711 (SMAC-mimetic) combined with standard regimes of chemotherapy	to evaluate dose escalation safety	Number of adverse events as a measure of safety and tolerability, anti tumor effect according to RECIST criteria
[182]	advanced or recurrent CRC	HLA-A24-restricted antigenic peptide, Survivin-2B80-88 vaccine, IFA and type 1 interferon (IFNalpha)	to assess whether immunogenicity of the Survivin-peptide could be enhanced with other vaccination protocols	Survivin-2B80-88 plus IFA and IFNalpha resulted in clinical improvement and enhanced levels of peptide-specific cytotoxic T lymphocytes
NCT02890069	CRC, non-small cell lung, renal cancer, triple negative breast cancer	PDR001 (anti-PD1 antibody), LCL161 (SMAC-mimetic), everolimus (mTOR inhib.), Panobinostat (histone deacetylase inhib.), QBM076 (CXCR2 antagonist), HDM201 (p53-MDM2 interaction inhib.)	to identify the doses and schedule for combination therapy and to assess the safety, tolerability, pharmacological and clinical activity of combinations	Dose limiting toxicities, frequency of dose interruptions and reductions, adverse and serious adverse events, changes in laboratory parameters, PFS
NCT00197912 NCT00978913	HLA-A2 positive, advanced melanoma and breast cancer	p53, Survivin and human telomerase peptide-pulsed dendritic cells	to show if autologous dendritic cells pulsed with peptides or tumor lysates can induce an immune response and clinical effects	Tolerability and safety, evaluation of treatment induced immune response and clinical tumor response/duration

Abbreviations: CRC, colorectal cancer; HLA, human leukocyte antigen; HLA-A2, Human leukocyte antigen serotype α2 domain; IFA, incomplete Freund’s adjuvant; MTD, maximal tolerable dose; OS, overall survival; PFS, progression–free survival; RECIST, Response Evaluation Criteria in Solid Tumors.

## Data Availability

No novel data were created in this study. Data sharing is not applicable to this article.

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
