# Peer review of "Tumor Suppressor Protein p53 and Inhibitor of Apoptosis Proteins in Colorectal Cancer—A Promising Signaling Network for Therapeutic Interventions"

_cancers, 2021, doi:10.3390/cancers13040624_

Round 1
Reviewer 1 Report
Güllülü et al. have summarized P53 dependent IAPs regulations and its implications in CRC. Manuscript shows up to date molecular information of the IAPs in CRC. Manuscript is written well. Figures are nicely presented and well made and easy to follow. Citations are up tp date. Authors have attempted to explore clinical aspects of IAPs. I have following suggestions which make this review more significant.
- I would suggest to include clinical trial information in a table format.
- It would be more helpful if authors should include a paragraph regarding challenges in clinical settings related to IAPs use.
- Addition of preclinical data in a table format would further signify the exploitations of IAPs in CRC.
- I would suggest to include more clinical information in "Clinical treatment potential by IAPs and p53" section.
- It would also be more helpful if authors would include a figure related to IAPs treatment strategies in CRC.
Author Response
Point-to-point answers to Reviewer # 1
- I would suggest to include clinical trial information in a table format.
According to the reviewer´s kind suggestion, we now have included a novel Table 2 summarizing clinical trial information and supplemented the text accordingly.
- It would be more helpful if authors should include a paragraph regarding challenges in clinical settings related to IAPs use.
We would like to thank the reviewer for his/her valuable comment. In a revised version of section “Clinical treatment potential by IAPs and p53”, we now have included a paragraph on challenges in clinical use containing adverse effects like a cytokine storm (lines 519-527).
- Addition of preclinical data in a table format would further signify the exploitations of IAPs in CRC.
To increase readability of the manuscript, we now provided recent preclinical studies targeting IAPs either alone or in combination with p53 in CRC in a novel Table 1.
- I would suggest to include more clinical information in "Clinical treatment potential by IAPs and p53" section.
In a revised version of the section “Clinical treatment potential by IAPs and p53”, we now included clinical informations in more detail by attaching a second novel Table 2 that summarizes current clinical phase I/phase II trials.
- It would also be more helpful if authors would include a figure related to IAPs treatment strategies in CRC.
Again, we would like to thank the reviewer for his/her insightful suggestion. In the revised version of our manuscript, we now have included a novel Figure 3 related to therapeutical approaches that target IAP expression in CRC.
Reviewer 2 Report
Reviewers’ Comments (Remarks to the Authors):
This manuscript reviews the molecular regulation mechanisms between IAPs (inhibitor of apoptosis protein family) and tumor suppressor p53 and discuss in detail the therapeutic potential of targeting their interrelationships by multimodal treatment options.
The authors describe the different components of the IAPs family and determine their functional relationship with p53, as a potential way to develop therapies aimed at the treatment of colorectal cancer. In a detailed way and through a clear grammar, the authors encourage a reading by the reader, making the review attractive, entertaining and with a great scientific value; being necessary characteristics to attract readers to the MDPI group.
Minor considerations :
- The manuscript is well written, however as minor comment I recommend doing a revision of the English language to avoid the repetition of terms to describe the function of AIPs and p53, cell pathways (DNA repair of course) or their expression levels, such as "increased" or "decreased", and use / combine more as "triggers" or "promotes". However, I reiterate that the grammatical level of the text is adequate.
- As a minor comment and without the intention of doubting the opinion of the corresponding author, I suggest “rewriting” (in case to be considered) the title of your review to make it more impressive and attract with greater force future readers of Cancers journal.
- The introduction about CRC is appropiated but I suggest the authors rewrite sentences 50 to 58 introducing the sequence of events that lead to the malignancy of the epithelium until reaching the metastatic CRC phenotype; It is not necessary to extend the introduction any further, only to highlight this idea. As a guide I attach a possible "sequence of events » :
APC mut k-RAS P53
Normal epithelia…. dysplasic epithelia…. adenoma…. carcinoma…. matastasic CRC
- PARP1 is considered the main genome guardian. According the information detailed in SECTION 2 « Biology and fucntions of p53, a brief introduction » Could the authors briefly explain the relationship with p53 (in 2 sentences maximum)? Is there a PARP1 / p53 relationship with CRC progression? Do the authors know if PARPs inhibitors are used in CRC clinic? It is not necessary to include such information in the review unless there are strong clinical data relating to both proteins and the progression of CRC.
Major considerations :
As a general and important comment the main consideration that I present in my review of this work is going to focus on SECTION 3 and 4: The authors describe in detail all the members of the AIP family, with structure, functions; However, I consider that this section is sufficiently described in the literature. I propose that the authors consider reducing section 3 and expanding section 4. I believe that section 4 could be completed with information from section 3, including more bibliography with clinical and basic data; so that the authors can redesign figure 1 including data from section 4, which I consider has a very important weight to make the review attractive to future readers
Author Response
Point to point answers to Reviewer # 2
- As a general and important comment the main consideration that I present in my review of this work is going to focus on SECTION 3 and 4: The authors describe in detail all the members of the AIP family, with structure, functions; However, I consider that this section is sufficiently described in the literature. I propose that the authors consider reducing section 3 and expanding section 4. I believe that section 4 could be completed with information from section 3, including more bibliography with clinical and basic data; so that the authors can redesign figure 1 including data from section 4, which I consider has a very important weight to make the review attractive to future readers.
We would like to thank the reviewer for his/her helpful proposal. In a revised version of section 4, we now have expanded the text including more detailed clinical and preclinical data, while we aimed to shorten section 3. Moreover, we redesigned Figure 1 by including effects of IAPs on radio- and chemoresistance and their importance as prognostic and predictive biomarkers in CRC.
- The manuscript is well written, however as minor comment I recommend doing a revision of the English language to avoid the repetition of terms to describe the function of AIPs and p53, cell pathways (DNA repair of course) or their expression levels, such as "increased" or "decreased", and use / combine more as "triggers" or "promotes". However, I reiterate that the grammatical level of the text is adequate.
Throughout the entire manuscript, we have exchanged/combined “increased” by “triggers” or “promotes”.
- As a minor comment and without the intention of doubting the opinion of the corresponding author, I suggest “rewriting” (in case to be considered) the title of your review to make it more impressive and attract with greater force future readers of Cancers journal.
According to the reviewer´s kind suggestion, we now changed the title to: “Tumor suppressor protein p53 and inhibitor of apoptosis proteins in colorectal cancer – a promising signaling network for therapeutic interventions”.
- The introduction about CRC is appropiated but I suggest the authors rewrite sentences 50 to 58 introducing the sequence of events that lead to the malignancy of the epithelium until reaching the metastatic CRC phenotype; It is not necessary to extend the introduction any further, only to highlight this idea. As a guide I attach a possible "sequence of events » : APC mut k-RAS P53 Normal epithelia…. dysplasic epithelia…. adenoma…. carcinoma…. metastasic CRC.
Again, we thank the reviewer for his/her valuable comment. In a revised version of the introduction section, we now in more detail describe the sequence, histological changes and genomic alterations associated with the adenoma-carcinoma sequence in CRC (lines 50-60).
- PARP1 is considered the main genome guardian. According the information detailed in SECTION 2 « Biology and fucntions of p53, a brief introduction » Could the authors briefly explain the relationship with p53 (in 2 sentences maximum)? Is there a PARP1 / p53 relationship with CRC progression? Do the authors know if PARPs inhibitors are used in CRC clinic? It is not necessary to include such information in the review unless there are strong clinical data relating to both proteins and the progression of CRC.
In line with the reviewer´s insightful objection, in a revised version of section 2, we now briefly describe a relationship between p53 and PARP-1 (lines 88-92). In addition, not to exceed the volume of our manuscript and in line with recent data on little benefit and high toxicity of PARP inhibition in colorectal cancer, we decided to omit clinical findings on PARP inhibition.